# Improving understanding of disease control implementation research through a mooc with participants from low- and middle-income countries: Evaluating participant reactions and learning

**Adanna Nwameme**[1], **Phyllis Dako-Gyeke**[1], **Emmanuel Asampong**[1], **Pascale Allotey**[2], **Daniel D. Reidpath**[3], **Edith Certain**[4], **Mahnaz Vahedi**[5], **Bella Ross**[6], **Dermot Maher**[5], **Pascal Launois**[5]*

1 Department of Social and Behavioural Sciences, School of Public Health, University of Ghana, Legon, Ghana, 2 United Nations University—International Institute for Global Health (UNU-IGIH), UKM Medical Centre, Jalan Yaacob Latif, Bandar Tun Razak, Federal Territory of Kuala Lumpur, Malaysia, 3 Health Systems and Population Studies Division, Mohakhali, Dhaka, Bangladesh, 4 Consultant, Special Programme for Research and Training in Tropical Diseases (TDR), WHO, Geneva, Switzerland, 5 Special Programme for Research and Training in Tropical Diseases (TDR), WHO, Geneva, Switzerland, 6 Learning Design and Digital Innovation, Monash College, Docklands, Victoria, Australia

* launoisp@who.int

**Data Availability Statement:** Data is available through the repository location: Ross, B. (2023,

## Abstract

The Special Programme for Research and Training in Tropical Diseases developed a massive open online course (MOOC) on implementation research with a focus on infectious diseases of poverty (IDPs) to reinforce the explanation of implementation research concepts through real case studies. The target MOOC participant group included public health officers, researchers and students. By reshaping institutions and building resilience in communities and systems, implementation research will allow progress towards universal health coverage and sustainable development goals. This study evaluates learners' knowledge in implementation research after completing the MOOC using anonymous exit survey responses. Of the almost 4000 enrolled in the two sessions of the MOOC in 2018, about 30% completed all five modules and the assessments, and were awarded certificates. The majority of the participants were early to mid-career professionals, under the age of 40, and from low- and middle-income countries. They were slightly more likely to be men (56%) with a Bachelor or a Master's degree. Participants were public health researchers (45%), public health officers (11%) or students (11%). On completion of the course, an exit survey revealed that 80.9% of respondents indicated significant improvement to strong and very strong implementation research knowledge. This evaluation clearly shows the usefulness of the MOOC on implementation research for reaching out to field researchers and public health practitioners who are facing problems in the implementation of control programmes in low- and middle-income countries.

March 15). Implementation Research MOOC.
https://doi.org/10.17605/OSF.IO/8YSP9.

**Funding:** The author(s) received no specific
funding for this work.

**Competing interests:** The authors have declared
that no competing interests exist.

## Author summary

Implementation research (IR) focuses on how to improve access to, and distribution and
use of disease control interventions, which are under-utilised in many low- and middle-
income countries. Effective implementation of disease control interventions can have sig-
nificant impacts by reducing illness and mortality. Despite this, IR is not widely recog-
nised, researched or taught. Education and training in IR is not readily available or
accessible and, for this reason, Special Programme for Research and Training in Tropical
Diseases (TDR) developed a massive open online course (MOOC) on the topic of IR with
a focus on infectious diseases of poverty (IDPs). A MOOC is a free online course that any-
one can participate in and can provide training and education to large audiences of learn-
ers. A MOOC may consist of online media, short videos, lectures, case studies, quizzes and
other assessments, online discussion forums, and readings. The TDR IR MOOC targeted
field researchers and public health practitioners facing problems in the implementation of
control programmes in low- and middle-income countries and was open to anyone who
was interested. In this study, we explore the demographics of those who registered for, par-
ticipated in, and completed the MOOC, and whether the MOOC increased their knowl-
edge of the topic. Findings reveal that registrants were most commonly from the WHO
African region, were English language speakers and public health researchers. The MOOC
was found to be successful in increasing learners' knowledge in implementation research.

## Introduction

Many inexpensive and effective disease control interventions remain under-utilised in many
low- and middle-income countries (LMICs). This under-utilisation is linked to the fact that
these control tools may behave differently in the field compared with their performance in
controlled trials or laboratory settings. Research on how to improve access, distribution and
use of these tools, i.e. implementation research (IR), can make a critical difference to health
outcomes.

Although there is a lack of infrastructure, weak functioning health systems, and a reduced
number of health care workers in LMICs, bringing innovation to the field requires the same
level of scientific rigour that is used for the development of the innovation. IR is a critical tool
for providing scientific evidence to improve public health programmes. To respond to the
need for such scientific rigour, Special Programme for Research and Training in Tropical Dis-
eases (TDR) developed a continuum of short training courses on IR including the TDR toolkit
on IR (http://adphealth.org/irtoolkit/); a short in-person training course on basic principles in
IR [1]; a training course on ethics in IR [2]; and the MOOC on IR with a focus on infectious
diseases of poverty (IDPs) [3].

The TDR is a pioneer in IR, particularly in the control of malaria and onchocerciasis (river
blindness). Home and community-based management of malaria became key elements in
TDR's programme during the 1998–2006 period and beyond. This involves the training of
mothers, drug vendors, village volunteers and teachers in the first line of care for malaria
where health clinics and health care providers are not accessible. The effectiveness of home
management has been demonstrated by TDR to reduce mortality by at least 40% in some stud-
ies[4,5]. The elimination of river blindness as a significant public health problem was first
achieved in 11 West African countries due to a combination of approaches including the use
of vector control tools, diagnosis and mass drug treatment with ivermectin. Building on these

efforts, community-directed treatment with ivermectin was used by the African Programme for Onchocerciasis Control (APOC) to eliminate river blindness in 19 additional African countries [6,7].

Despite the success of IR, the concept of IR remains relatively new and undervalued in research. Also, although some universities are beginning to include IR in their training curriculum (e.g. TDR supports seven universities to offer a Master's degree with a focus on IR)[8], IR research is generally not taught in academic institutions. This lack of recognition has impaired the development of careers in IR for young scientists. The key challenge for universities to include IR in their courses is that IR requires multidisciplinary and inter-disciplinary approaches spanning across both public and private sectors. Many access and delivery issues that hinder adoption and scaling up of an intervention are not viewed as problematic by policymakers. Thus, IR is not valued by disease control managers and decision-makers [9]. There is, therefore, a need to develop specific methodologies with a double objective to improve the development and use of knowledge and skills in IR by researchers; and to demonstrate the value of implementation research concepts to policymakers [10]. Targeting these distinct groups separately may lead to increased uptake in IR.

A massive open online course (MOOC) provides a new way of delivering training and education to a large audience of learners through online media, including short videos of formal presentations, lectures or case studies, complemented by recommended readings, moderated discussion forums and automated assessments. In 2019, it was estimated that 110 million learners participated in around 13500 MOOCs developed by over 900 universities [11]. MOOCs are scalable to many participants because they only need internet connectivity for delivery. In the absence of subscription fees, anyone with internet connectivity and an interest in the topic can enrol in a MOOC to access course resources, with opportunities to interact and share their knowledge with peers, thus making education more accessible to a massive audience.

This efficient delivery medium could potentially increase the efficacy of public health interventions that are being managed by field researchers and public health practitioners. For these reasons, TDR decided to develop a MOOC on IR with a focus on IDPs to illustrate IR concepts using real case studies.

## The TDR MOOC on IR with a focus on IDPs

This MOOC was designed as a step-by-step training programme for public health researchers and decision-makers, disease control programme managers, academics and others focusing on designing robust IR projects to make proven health interventions more widely available to people at risk of diseases of poverty. The course was centered on the control of malaria and onchocerciasis and it was presented and interpreted by experienced public health researchers, practitioners and academics. The objective was to educate participants about what IR is and how it can be used in practice.

The development of the TDR MOOC on IR with a focus on IDPs took a bottom-up approach. After an extensive review of the literature on IR, TDR identified members of a MOOC working group to attend an initial workshop to agree on the modular structure and the curriculum of the MOOC. Each module of the MOOC was then assigned to a particular expert for content development. Among the 16 experts invited, ten came from LMICs (Botswana, China, Colombia, Ghana, India, Indonesia, Kazakhstan, Kenya, Philippines and Tunisia) from each of the WHO regions and six came from high-income countries (Germany, Portugal, Switzerland, UK and USA).

During the aforementioned workshop, the working group agreed that the MOOC would comprise step-by-step online videos and interactive training activities (with subtitles in

English, French and Spanish). The MOOC consists of five training modules given over eight to nine weeks. The modules and contents are as follows:

- Module 1: The definition of IR and the assessment of the appropriateness of existing disease control programmes

- Module 2: The identification of challenges in various health settings

- Modules 2 and 3: The development of new interventions and strategies by working with communities and stakeholders

- Module 3: The specification of IR questions and the design of rigorous research projects

- Module 4: The identification of IR outcomes and evaluation of implementation effectiveness

- Module 5: Plans for scaling up implementation strategies in real-life settings

A second workshop was organised with the content experts to review and harmonise the contents of each module and identify potential MOOC presenters for various topics based on their ability to speak confidently in public while delivering lectures and presentations. Finally, the contents of the MOOC were reviewed by two external independent reviewers [12].

The MOOC was hosted and run on the edX platform and offered by TDR. A pilot study was set up at the end of 2017 in which 110 participants were invited to take part. Their feedback was then used to refine aspects of the contents and delivery, which included timing and workload, and issues with discussion forum engagement and language [12]. Such findings are in line with other studies regarding issues with discussion forums [13,14,15,16].

Two sessions of the final version of the TDR MOOC on IR were organised in 2018; the first from May to July and the second from October to December. Each learning module lasted a week but remained open until the closing of the MOOC to allow late completion. Participation in the MOOC was open. Basic demographics information was obtained at the beginning of the training using Datacol, a WHO survey tool.

To obtain a certificate of completion, learners were required to pass four evaluations in the form of four quizzes (one for each of the first four modules) and submit the outline of an implementation research project at the end of module 5. For the research proposal, learners were asked to identify an implementation problem close to their own experience and prepare a short outline of a proposal. The proposals were required to include: 1) a statement of the implementation problem; 2) the definition of the implementation approach; 3) the development of a conceptual diagram; and 4) a list of the IR outcomes to be measured. Each proposal was peer-reviewed and graded by three MOOC participants with the final grade calculated based on the average of the reviews. The four quizzes represented 60% of the total final mark, while the proposal contributed to the remaining 40%. The pass mark to obtain a certificate of completion was set at 70% of the maximum possible score. This pass mark was chosen to balance the weight of the four quizzes and the final proposal.

## Research Aims

The primary aim of this research is to explore participants' understanding of IR following completion of the TDR MOOC, and their reaction to the MOOC using Kirkpatricks' method of evaluating a training programme [17]. In addition, this paper outlines the demographics of the MOOC participants, those who completed the MOOC as well as those who responded to the exit survey.

## Methods

### Ethics statement

Ethics approval was granted by the Monash University Human Research Ethics Committee (Project id: 26045). Respondents provided their written consent to take part in the research through their participation in the survey.

On completion of the MOOC, all participants, regardless of whether or not they had received a certificate of completion, were invited to complete an anonymous exit survey using Datacol. The anonymous exit survey investigated participants' experiences of the MOOC and their perceptions of learning. The exit survey collected quantitative and qualitative data and consisted of 29 open-ended, multiple choice, and 5-point Likert-type questions (ranging from 'to a very small extent' to 'to a very large extent', 'poor' to 'excellent', 'none' to 'very strong'). The survey captured participants' demographics, expectations of, satisfaction with and evaluation of the MOOC, and understanding of IR before and after completion of the MOOC. Participants were asked to assess how much their knowledge of IR had improved using a five-point scale to self-assess both their prior and current knowledge. The following scale was used: 0: none, 1: weak, 2: moderate, 3: strong and 4: very strong IR knowledge before and after the MOOC.

The data reported on here is therefore self-reported by participants following completion of the MOOC. The survey aimed to evaluate whether participants' knowledge in IR had improved after taking part in the MOOC, and how they rated the MOOC. The findings from the exit survey are presented using descriptive statistics.

The MOOC was evaluated using Kirkpatricks' four-level model [17]. This model is the most widely used method for evaluating training programmes and targets four levels: 1) Reaction–the way learners react to the experience of the course; 2) Learning–the new knowledge, skills and attitudes they gained during the course; 3) Behaviour–how the new knowledge, skills and attitudes are applied; and 4) Results—improved job and organisational performance. This paper analyses the first two steps.

Other data reported on here include descriptive statistics of the demographics of MOOC enrolments gathered using a survey at the beginning of the MOOC. Descriptive statistics were calculated using Microsoft Excel. The demographic information gathered of MOOC participants included country of residence, language, education level, gender and age.

## Results

### MOOC participation

There were 3858 registrants enrolled in the two sessions of the MOOC. 2470 (64%) completed the first quiz; 1944 (50.4%) the second; 1832 (47.5%) the third; and 1304 (33.8%) completed the fourth quiz. At the end of the MOOC sessions, 1163 (30.2% of the total initially enrolled and 89.2% of those who completed the course) met all the requirements for the course including submission of the final project outline, achieved the mandatory pass mark, and received a certificate of completion.

### Overview of demographics

The following Table 1 shows a comparison of key demographic features of those who enrolled in the IR MOOC, those who completed the MOOC and obtained a certificate of completion, and those who responded to the online survey. Information is presented in order of most common categories first.

**Table 1. A comparison of MOOC participants', completers' and survey respondents' demographic profiles.**

| Demographic parameters | MOOC registrants (N = 3858) | MOOC completers with demographic information (N = 546) | Survey participants (N = 574) |
|---|---|---|---|
| Sex | Men (56%)<br>Women (44%) | Men (53%)<br>Women (47%) | Men (58%)<br>Women (42%) |
| Age | 20–40 years old (77.5%) | 20–40 years old (84.5%) | |
| WHO region | African region (62.4%)<br>South-East Asian region (17.7%)<br>European region (2.8%)<br>Americas region (9.9%)<br>Western Pacific region (2.2%)<br>Eastern Mediterranean region (5.5%) | African region (59.3%)<br>South-East Asian region (28.8%)<br>European region (1.6%)<br>Americas region (7.5%)<br>Western Pacific region (2.5%)<br>Eastern Mediterranean region (0.1%) | African region (67.5%)<br>South-East Asian region (9.9%)<br>European region (9%) |
| Profession | Public health researchers (45%)<br>Public health officers (15.5%)<br>General practitioners (11.1%)<br>Students (11%) | Public health researchers (53.1%)<br>Public health officers (11.7%)<br>General practitioners (5.5%)<br>Students (17.3%) | Public health researchers (36.8%)<br>Public health officers (10.7%)<br>General practitioners (13%)<br>Students (16.4%) |

## Demographics of enrolled learners

Registrants came from 115 different countries, See Fig 1. The most represented LMICs with more than 1% of registrants were Nigeria (1191/30.9%); India (373/9.7%); Uganda (192/5%); Ghana (155/4%); Kenya (143/3.7%); Nepal (141/3.7%); Cameroon (124/3.2%); Malawi (121/3.1%); Sudan (106/3.1%); Ethiopia (91/2.4%); Myanmar (66/1.7%); Colombia (61/1.6%); Honduras (57/1.5%); Rwanda (53/1.4%); Tanzania (51/1.3%); The Philippines (49/1.3%); Sri Lanka (42/1.1%); and Zambia (39/1%).

The majority of the registrants were from the WHO African region. Those from the WHO South-East Asian region were mainly from India (56%). Apart from India, which represented 9.7% of the registrants, very few registrants were from the other BRICS (Brazil, Russia, India,

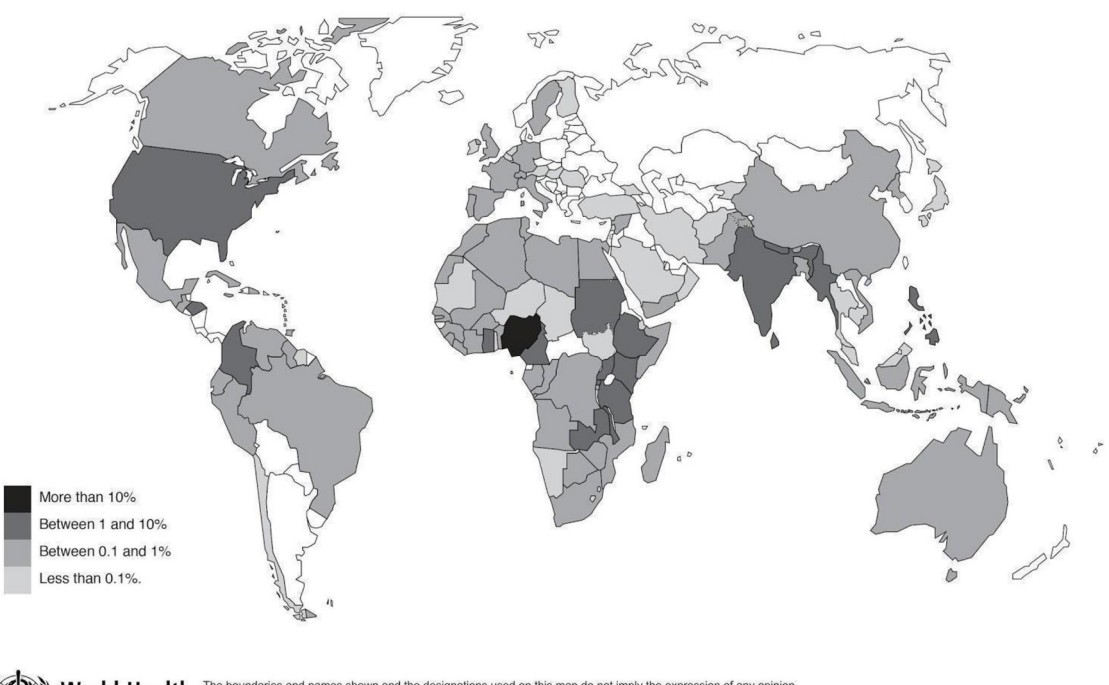

**Fig 1. Represented countries of MOOC registrants.**

China and South Africa) countries: 0.5% for South Africa, 0.4% for Brazil, and 0.2% for China. The Russian Federation was not represented at all. The majority of registrants came from English speaking countries (87.2%) followed by Spanish (7.7%), French (4.6%) and Portuguese speaking countries (1.4%). Note that the total is slightly higher than 100% as some countries are considered bilingual (e.g. Cameroon and Rwanda) and it was not possible to identify the countries of origin.

Men were more commonly represented in the WHO African region (63.4% versus 36.6% for women); however, in the other WHO regions, women were equally or more commonly represented than men. In terms of age profile, the majority of registrants were between 20 and 40 years old. Women were slightly younger than men: 39.6% of women were under 30 years of age as compared with 31.8% of men.

The highest number of registrants had an MSc degree followed by a Bachelor degree, and a PhD. Six hundred and eighty (680) enrollees had a Medical Doctorate (MD). There were no differences in gender except for PhDs which were held by a higher number of women (55.7% versus 44.2% for men). Registrants were mainly public health researchers followed by public health officers, general practitioners and students. Women were more often public health officers and students than men (51.6% versus 46.5% and 56.8% versus 43.3%, respectively). Women were less often public health researchers (41% for women versus 59% for men) and general practitioners (43% for women versus 57% for men).

## Demographics of learners who completed the course

In total, TDR obtained demographic information for 546 learners who completed the MOOC. Completers, who passed the course came from 56 different countries. The most represented LMICs with more than 1% representation were Nigeria (108/19.8%), India (92/16.8%), Uganda (52/9.5%), Nepal (38/7%), Ghana (37/6.7%), Kenya (24/4.4%), Cameroon (20/3.7%), Malawi (18/3.3%), Myanmar (12/2.2%), Rwanda (12/2.2%), Philippines (11/2%), Ethiopia (9/ 1.6%), Sri Lanka (8/1.5%), Tanzania (7/1.3%), Honduras (5/0.9%), Zambia (5/0.9%), Zimbabwe (5/0.9%), Ecuador (4/0.7%), Burkina Faso (4/0.7%), and Democratic Republic of the Congo (4/0.7%).

The majority of the completers came from the WHO African region followed by the WHO South-East Asian region (mainly from India and Nepal at 58.6% and 24.2% respectively), the WHO Americas region, the WHO Western Pacific region, the WHO European region and just a few from the WHO Eastern Mediterranean region. Apart from India which represented 16.8% of the completers, the number of completers was very low for the other BRICS countries: 0.1% for South Africa, 0.3% for Brazil and 0.1% for China. There was no participant from The Russian Federation. A majority of the completers came from English speaking countries (86.2%) followed by French (9.3%), Spanish (4.2%) and Portuguese speaking countries (0.3%).

Globally, just over half of all completers were men. Men were more represented (60.6%) than women (39.6%) in the WHO African region, although women were equally or more represented than men in the other WHO regions. In terms of age profile, the majority of completers were between 20 and 40 years old. Women were, on average, slightly younger than men, with 34.7% of women under 30 years of age as compared to 38.7% of men.

The highest number of participants who completed had an MSc degree followed by those with a Bachelor degree and a PhD. Sixty-two completers held a Medical Doctorate (MD). Completers were mainly public health researchers followed by students, public health officers and general practitioners. Interestingly, women were less often public health researchers than men (28.9% versus 71.4%) but were more often students than men (58.7% and 13.2%, respectively).

### Results of the MOOC exit survey

An anonymous exit survey was sent at the end of each TDR MOOC session to identify potential improvements in the registrants' IR knowledge. Of the total enrolments (3858), 574 responded to the survey (responders) representing a response rate of approximately 15%. Responders came from 52 countries, and the ten most-represented countries were identical to those found at the registration phase except for an unexpectedly high response rate from learners in Myanmar.

The majority of responders were from the WHO African region, reflecting a similar proportion to those of the registrants. Compared with the registrants, there was a higher representation of responders from the WHO European region and less representation from the WHO South-East Asian region. 58% of responders were men, and 42% were women. They were public health researchers, public health officers, general practitioners and students. Altogether these results showed that the population of responders was similar to the population of registrants.

### Reaction of the learners to their experience of the course

Of the responders, 72.3% indicated that, to a large extent, the MOOC met their expectations. The expectations of the responders were not at all met for only 0.2%, to some extent for 9.3% and to a moderate extent for 18.2%. The vast majority of responders rated the MOOC as excellent (52.2%) or very good (40.7%). Among the different components of the MOOC, the videos, readings and the assessments, including quizzes, contributed the most to the learning. The discussion forum was identified as the element that contributed the least to the learning, and it was evident that registrants underutilised this interactive opportunity.

### Factors influencing the receipt of a certificate of completion

Among the 574 responders, 438 obtained a final certificate of completion and 136 did not. The number of women was lower in the group who got a certificate than in the group who did not get a certificate (39.8% versus 47.4%).

The responders who received a certificate and those who did not are similar in terms of education level as they were mainly Master's degree holders (53.6% of those who received a certificate versus 51.1% of those who did not). In terms of age, there was little difference between those that obtained a certificate and those that did not. 69.3% of responders who received a certificate were less than 40 years old while 72.5% of responders who did not qualify for a certificate were in the same age group.

There were no differences in responders' familiarity with the use of online training programs and whether or not they received a certificate. The number of online courses previously taken by responders who did or did not obtain a certificate was similar. Results show that 44.2% of responders received a certificate, and 47.4% of responders never participated in a MOOC.

There was no marked difference in the length of field experience between the responders who received a certificate and those who did not. Of those who worked for less than ten years in the field, 72.8% of responders obtained a certificate, and 68.1% did not. Also, of those who worked between 10 and 20 years in the field, 21% of responders received a certificate in comparison to 26.7% of those who did not.

The only difference found between the responders who did and did not obtain a certificate was with their occupations. Public health researchers were more likely to receive a certificate than not (45.2% versus 41%), while public health officers were less likely to receive a certificate

than not (10.7% versus 19.7%). Students were also more represented in the group of responders who received a certificate as compared to the group who did not (16.1% versus 10.6%).

## Improvement of the understanding of IR

On completion of the MOOC, participants were asked to assess how much their knowledge of IR had improved using a five-point scale (from 'none' to 'very strong') to self-assess both their prior and current knowledge. On average, survey respondents' knowledge increased by approximately 2 points, equivalent to a shift from 1.1 (weak knowledge) at the beginning of the course to 3 (strong knowledge) on completion of the course. There was no difference found between gender. The mean knowledge increased by 1.1 (weak) before the course to 3.1 (strong) after the course for men and from 1.1 (weak) before the training to 3.4 (strong) after the course for women.

Public health researchers' knowledge increased by approximately 2 points on average, equivalent to a shift from 1.1 (weak knowledge) at the beginning of the course to 3.2 (strong knowledge) on completion of the course. The self-reported knowledge increase was slightly less for public health officers and students with an increase from 1.1 at the beginning of the course to 2.9 on completion of the course. Regardless of occupation, there were no gender differences.

In all the groups tested (i.e. public health researchers, public health officers and students), the number of responders who indicated an improvement in IR knowledge was consistently higher in the group of responders who obtained a certificate than in the group of responders who did not. A majority of the public health researchers (84.3%) who received a certificate indicated strong/very strong knowledge after the MOOC as compared to 66% of the public health researchers who did not receive a certificate. Similarly, 81.8% of the public health officers who obtained a certificate indicated strong/very strong IR knowledge after the course as compared to 43.9% of the public health officers who did not. Finally, 88% of the students who obtained a certificate showed strong/very strong knowledge in IR after the MOOC as compared to 64% of the public health researchers who did not obtain a certificate.

## Discussion

The MOOC attracted learners from a wide range of backgrounds. Registrants came from over one hundred countries from the six WHO geographical regions with most from the WHO African region. The majority of the registrants were from anglophone countries with significantly lower numbers from Spanish, French and Portuguese speaking countries. The demographics of those who completed the course reflected those of the registrants. Apart from India, there was very little representation from the BRICS countries. Feedback from the Spanish speaking registrants from South America clearly showed that English was the dominant language of the MOOC, as it is in health research in general, and overshadowed other languages thereby limiting other linguistic communities. Studies have shown that learners may be reluctant to join online discussions in a language in which they are not comfortable [18] and this was likewise found to be the case in the 2017 pilot of the IR MOOC [12]. In response to this issue, TDR decided to translate the MOOC into all six United Nations languages. The Spanish and French versions were tested in 2018, and the Russian and Chinese versions were made available in 2020. The Arabic version was made available at the beginning of 2021.

It was not surprising that health officers were less likely to receive the Certificate of Completion due to time constraints related to their professions. The comparatively high rate of completion was, however, unexpected given their limited time available to allocate to such training. In response to this finding, the MOOC designers considered the following options: 1)

Create a specific assessment for module 5. This option was deemed difficult to implement. 2) Create a team work component including researchers and implementation managers for all participants in the final module of the MOOC. This option has been piloted in the latest sessions of the MOOC but the results have not been obtained. 3) Develop a specific 1-hour module with a corresponding assessment for different learner cohorts. This option, identified through consultation conducted by the United Nations University International Institute for Global Health on training in IR, is one of their recommendations and will be implemented in future iterations of the MOOC.

The results presented in this paper clearly show that by using the MOOC on IR, TDR was able to reach the main partners involved in IR i.e. public health researchers (45% of the registrants) followed by public health officers (15.5% of the registrants) and general practitioners (11.1% of registrants). Students also represented 11% of the registrants, which is not surprising due to the well-known historical involvement of TDR research capacity strengthening activities in LMICs [19]. Those who completed the MOOC were very similar to the general participants in terms of their demographic profiles. Completers were mainly young men—either public health researchers or public health officers—from the WHO African region with a Master's or Bachelor degree. The only difference found was that the women who completed the course were more frequently students than men were.

The completion rate of this MOOC was 30% which is higher than the 5–10% completion rate observed in high-income countries [13,14,20]. A survey conducted in three countries (Colombia, The Philippines and South Africa) [21] identified a completion rate of between 30 and 48%—though it must be noted that the participants were all employed responders and used the MOOC for professional development purposes.

The high completion rate in the TDR MOOC could be because the content of the course was contextualised to local LMIC conditions and responded to a need. Indeed, the content had been developed by and for LMICs with this in mind and illustrated concepts with real-life studies from these countries involved. Tailoring content to local contexts has been identified as an important consideration for learners in the global South and one that may lead to higher retention rates and increase the success of MOOCs in LMICs [22]. Also, the 30% completion rate represents only the certified learners; many registered learners who, on occasion, viewed the uploaded videos to gain the skills they deemed necessary for their purposes should also be recognised. These learners are called browsers or auditors. In this MOOC, 64% of the enrolled participants watched at least one module.

The findings reveal that public health officers tend not to complete the course and obtain a certificate of completion. This contrasts with public health researchers and students for whom the certification may be critical for their career development (10.7% versus 19%) should they wish to pursue IR-related research, further study or employment. Perhaps there is no imperative need for public health officers to obtain professional certification. Public health officers could also be more likely to be browsers or auditors than active learners. Finally, public health officers generally do not have the research expertise to develop the proposal that is submitted at the end of module 5 as part of the requirements for completing the course. A quiz at the end of module 5 may be more suitable in order for public health officers to receive a certificate.

The fact that female students more often completed the course than males (58.8% versus 13.2%) may reflect the challenges that women generally face in their career development, such as being less likely to achieve the same level of career development and salary as men or face discrimination and constraining opportunities in leadership roles, that may be amplified in LMICs [23,24,25]. A certificate is likely more critical for women than men to make advancements in their careers.

In relation to participants' experiences of the TDR MOOC, it was found that a key strength of the MOOC is that it encourages and enables registrants to share their knowledge and their challenges with their peers through forum discussions. The finding that this MOOC was able to involve two main stakeholders (public health officers and researchers) is a very positive result, as this was the target audience for the MOOC. However, in the two sessions of the MOOC, the discussion forum was identified as the element that contributes the least to participants' learning. Studies reveal that discussion forums are often a weak link in MOOCs due to poor engagement, slow responses and the quality of the discussions [e.g. 13,14,15,16]. Thus, in response to this challenge, TDR developed a training course on creating, managing and facilitating online discussions. [For more on discussion forums, see 26,27].

A blended learning MOOC could be an effective way of delivering IR material. Some studies show that blending online course content with face-to-face interaction is beneficial for participants from LMICs [28], and could be a tool to improve the interaction between the different partners within IR. For this reason, TDR organised a blended programme in collaboration with the United Nations University, International Institute for Global Health and the Ministry of Health Malaysia's Institute for Health Systems Research. This programme was offered in 2018 with 77 participants in total: 12 participants from Monash University, 27 from the National Institutes of Health, 14 from The Ministry of Health of Malaysia Putrajaya, ten other academics and 14 individuals invited by TDR (managers of the MOOC in the Regional Training Centres supported by TDR from each WHO Region) [29]. For each module of the MOOC, participants were required to read or watch material in advance of a synchronous session. In these synchronous plenary sessions, experts presented and discussed various IR concepts. Separate sessions were also offered on specific subjects. Although limiting the MOOC's flexibility in terms of use and limiting the number of participants, feedback obtained from participants in this blended programme showed that it helped ground the learning material. In particular, participants stated that being able to use their local issues rendered the programme engaging. As such, blended programmes could be useful for specific institutions to improve the IR skills of their health workforce.

The results presented in this paper show an improvement in IR knowledge in each of the target groups. Globally, 80.9% of the participants indicated a strong or very strong improvement in IR knowledge after the MOOC. Still, at various levels: 68.7% of public health officers, 80.3% of public health researchers and 87.1% of students indicated a strong or very strong understanding of IR on completion of the course. TDR has developed a framework for core competencies in IR specifically for LMICs which comprises 11 domains, 59 competencies and 52 sub-competencies [30]. This instrument could be used to measure the specific improvement of knowledge in each competency before and after the MOOC training. The feasibility of using such a tool was demonstrated by assessing competency outcomes of the TDR MOOC training using another framework developed for a US-based training programme [31]. In addition to the effectiveness of the training programme, the framework could also be used to identify training needs and guide future training programmes on IR.

## Conclusions

The evaluation of the IR MOOC presented here used the four-level model developed by Kirkpatrick [17] that comprises four levels: 1) Reaction; 2) Learning; 3) Behaviour; and 4) Results. In this paper, we analysed the first two steps. An outcome of great importance is that the vast majority of registrants indicate strong or very strong IR knowledge at the end of the training demonstrating the usefulness of such an online training course for IR scientists in LMICs. These results show that the IR MOOC can indeed increase understanding of the topic. An

increase in IR knowledge may in turn strengthen IR capacity around noncommunicable diseases as evidenced through previous studies evaluating changes in professional behaviour [27] and organisations' professional results [32] suggesting real-world benefits to public health outcomes. The next step is to analyse the application of the IR skills gained by the registrants.

## Limitations

Several limitations to this study should be noted. Response bias may be present in those responding to the exit survey versus those who do not which may result in the positive results reported [33]. Validity issues may be present as the data is self-reported. Further, as survey respondents are highly self-selective, their responses may reflect autonomous motivations [34,35]. A final limitation is the lack of a control group due to the exploratory nature of the research.

## Acknowledgments

The authors would like to acknowledge the contribution of Ms Najoua Kachouri for her constant administrative support.

## Author Contributions

**Conceptualization:** Adanna Nwameme, Phyllis Dako-Gyeke, Emmanuel Asampong, Pascale Allotey, Daniel D. Reidpath, Edith Certain, Mahnaz Vahedi, Dermot Maher, Pascal Launois.

**Formal analysis:** Bella Ross, Pascal Launois.

**Writing – original draft:** Adanna Nwameme, Phyllis Dako-Gyeke, Emmanuel Asampong, Pascale Allotey, Daniel D. Reidpath, Edith Certain, Mahnaz Vahedi, Bella Ross, Dermot Maher, Pascal Launois.

**Writing – review & editing:** Adanna Nwameme, Phyllis Dako-Gyeke, Emmanuel Asampong, Bella Ross, Pascal Launois.

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
