## [Decision Letter · Decision Letter 0]

21 Jul 2022

Dear Dr Ross,

Thank you very much for submitting your manuscript "Improving understanding of disease control implementation research through a MOOC with participants from low- and middle-income countries" for consideration at PLOS Neglected Tropical Diseases. As with all papers reviewed by the journal, your manuscript was reviewed by members of the editorial board and by several independent reviewers. In light of the reviews (below this email), we would like to invite the resubmission of a significantly-revised version that takes into account the reviewers' comments. 

We cannot make any decision about publication until we have seen the revised manuscript and your response to the reviewers' comments. Your revised manuscript is also likely to be sent to reviewers for further evaluation.

Sincerely,

Alberto Novaes Ramos Jr

Academic Editor

Victoria Brookes

Section Editor

Reviewer's Responses to Questions

**Key Review Criteria Required for Acceptance?**

**Methods**

-Are the objectives of the study clearly articulated with a clear testable hypothesis stated?

-Is the study design appropriate to address the stated objectives?

-Is the population clearly described and appropriate for the hypothesis being tested?

-Is the sample size sufficient to ensure adequate power to address the hypothesis being tested?

-Were correct statistical analysis used to support conclusions?

-Are there concerns about ethical or regulatory requirements being met?

Reviewer #1: Accept. Authors should include the CORE COMPETENCIES for IR

Reviewer #2: The work should definitely made accessible to the wider audience of researchers, policy makers, and implementers in the PLoS NTD. The data is promising to show interesting results that can be used for the furthering of MOOCs and similar programmes.

The study design is appropriate, the population size sufficient in order to come up with the results. However, the methods section has shortcomings as it 

- lacks some of the information that is placed in the introduction

- does not describe what was being asked for in the exit survey – on which questions the results are being based on!

- the usage of datacol is not described

- the statistical analysis is not described

Other points:

- some results are shown in the methods section, whereas some of the methods are being delivered during the discussion of the results

- when it comes to improvement, is there any baseline information from the participants (before – after, improvement from X to Y, etc?)

- aspects mentioned in the results are missing in the methods (see below)

- methods need to be specific and clear, should in principle allow anyone to repeat the work, in this respect, for instance the sentence is both grammatically wrong, non understandable (lines 161-162). Also, the “need for ethics approval” cannot be “waived”.

- was there any free text in the survey from which the results could gain insight?

Throughout the text, it remains a bit vague whether there is reference to participants of the MOOC versus the survey. Altogether, it is difficult to address the review criteria given for the methods.

Reviewer #3: The study presents its objectives and is well designed, the population is clearly described within an appropriate sample, although there may be no testable hypothesis. It should probably be explained a little more about the role and functions o MOOCs' training courses in the global learning strategies in the public health sector. We already have evidence of the profile of health professionals who enroll in this learning activity, especially in the global South. Actually, in some countries, MOOCs' learning experiences are relatively minor. Presencial encounters and practice experiments are fundamental since the health system's organizations are quite different. So, expectations and knowledge perceptions vary a lot. The statistics descriptions are fine but explain or advance little, in qualitative terms, about what was achieved given what the goal pursued. It seems evident that registrants came majoritarian from countries where some IR experiments already have been taking place, which confers some bias to results. It should also be considered that the health themes involved in case studies can make difference in interest of enrolments: in which countries onchocerciasis and malaria are the most critical IDR? Again, IR is about methodologies but is crucially about diseases to the fought, the population involved, and the possible local strategies available. One question: who were the 110 participants that took part of the pilot study? All the comments above did not remove the merit of the MOOCs' experience described in the text.

**Results**

-Does the analysis presented match the analysis plan?

-Are the results clearly and completely presented?

-Are the figures (Tables, Images) of sufficient quality for clarity?

Reviewer #1: Accept. Result may be presented in tabular form, if possible

Reviewer #2: The results match the indicated analysis plan, however it is difficult to follow the line of information throughout the paragraph:

- the order of listing should be consistent rather consistent than random – PhD – MD – MSc – Bachelor. 

- the group of participants would need to be defined once clearly and should not be mixed up by the authors throughout the text

- a participation with more than 1% representation is an unclear criteria, in particular when the ones with 0.7% participation are also listed.

The results section would benefit from (a) table(s) showing the demographics more clearly, with the countries of origin of the participants on the y axis, and the categories i) participated, ii) completed, iii) passed, [and possibly iv) completed the exit survey] on the x axis.

This would allow the deletion of abundant redundancies, i.e. lines 217-235 are repetitions.

Lines 237-238 are identical with 154-157.

As it comes to the reactions of the learners, there is no indication of the rating options they had, and hence any conclusion is difficult to assess, in particular when it comes to “improvement” of their IR knowledge

Methodology is missing for the following points that are outlined in the results and discussion:

- the “discussion forum” remains entirely unexplained hence is being shown as a result

- responders with and without certificate (271-275)

- occupations of the responders (276-280), how was this analysed?

- health officers (281 ff) – how was the grouping done (self-assessment?), which options were there and how was the distribution (see suggestion table)

- how was the “knowledge” or the “increase in knowledge” assessed? What does it mean “showed a strong/very strong knowledge in IR” – how was it measured, who assessed it, what is the difference between strong and very strong?

- what is referred to with “blending online course”? (385)

- what does this mean “helped ground the learning material”? and which “specific institutions” are meant (398 f.)

- not sure whether “globally” (403) conclusions can be drawn from 

- the reader asks the question in line 408 why the data are not being “used to measure the specific improvement for knowledge in each competency before and after the MOOC training” – the title and abstract implied this was (intended to be) done?

Some of the points should be shifted from the results to the discussion section (i.e. 286-291), whereas some of the discussion should be place to the results (i.e. 334-337).

Others should be shifted to the introduction, i.e. 375-376, 378-380.

Reviewer #3: As said earlier, results are mainly quantitative and adequately presented. Using more maps and some tables/figures should improve the understanding of results. Yet, if the focus was on those who completed an exit survey, those data could gain more emphasis. Since social science researchers are critical members of IR teams, this profile and gender issues could be more explored (see line 233). It is unclear to me the text in lines 281-292. The authors had the intention to promote any modifications during the course time? Both health researchers and managers/officers have time limitations to accomplish the course tasks. Moreover, I am not sure if public health researchers consider MOOC certification crucial for career development.

**Conclusions**

-Are the conclusions supported by the data presented?

-Are the limitations of analysis clearly described?

-Do the authors discuss how these data can be helpful to advance our understanding of the topic under study?

-Is public health relevance addressed?

Reviewer #1: Authors have made a novel effort in expanding knowledge through MOOC based training in a new and highly important area of Implementation Research for Public Health Programs. Very much needed for improving effectiveness of health programs in changing health outcomes and achieving sustainable development goal 3 - Universal Health Coverage.

Reviewer #2: The conclusions should include

- that it is a natural explanation that the majority of participants “came from EN speaking countries” since the MOOC is in EN language! It should also be mentioned by the authors that the MOOC offer in other languages is provided with subtitles in those languages only but the videos are all in EN language, which may be a reason for many participants not completing the course.

- the last section of the results section

It is unlikely that participants belong to a linguistic community (which is a homogenous group of people bound together by a common language) only when they come from same language speaking countries.

The discussion refers to demographic profiles which are not explained before, 339-342 is a repetition. There is sudden mention of “employed responders” which brings in an entirely new dimension in the analysis not mentioned before. Also, for which is the certification “critical for their career development” – was that assessed? Why is this stated to be more critical for women than for men? Who are “active learners”? As opposed to “public health officers”??

The text of the conclusion, however, is mainly discussion. The 4 level method was never mentioned before and cannot be part of the conclusion.

Same holds true for the introduction: “Many interventions remain under-utilised in many LMICs” – this is highly imprecise and subject to discussion in terms of correctness. In the beginning, IR would need to be explained for the reader.

The Introduction should be set up to introduce the topic discussed in the paper to the reader, it somehow starts from somewhere and goes to somewhere else. Decision-makers and researchers are being mixed up within one sentence although they would need to be looked at very differently

The reference for the MOOC on IR with focus on IDPs is missing.

There are definitely too many redundancies to present this text to the audience of PLoS NTDs, starting from the introduction and carrying through to the end of the text. As a consequence, the Intro is way too long and parts of it belong to the Methods section.

The goal of the study, however, is not described except with one sentence under the header “Research Aims”. This needs to be restructured and expanded. 

The authors could state more clearly and specifically how these results help advance the understanding of the topic and will have public health relevance.

The limitations would need more detailed information, including some explanations of the lack of a control group, the bias of the responders and its influence on the results, also see aspects above.

Reviewer #3: Much that was registered earlier fits for the Conclusion item. This piece is wellcome and can allow us to consider how to proceed to bring about IR to our empirical research on IDP. The text's title is "Improving understanding of disease control implementation research..." raises some doubts. The authors could make clear some points. Since we agree that IR is a multifaceted process, which specific topics were more important to explore? Saying differently, what was the main goal of the course? To introduce IR as a research strategy to reach better results on disease control? Or to go further in some topics helping researchers to design your own approach? Is it possible to answer what students can do now that we're not in the past? These are just some provocations to the authors.

**Editorial and Data Presentation Modifications?**

Reviewer #1: Highly recommended for publication. 

Include core competencies for Implementation Research

Data may also be presented in tabular form, if possible

Reviewer #2: Please see above for suggestions on

- a table

- shifting text elements between the headers introduction - methods - results - discussion - conclusion

- delete redundancies

Reviewer #3: Maybe the authors could change the title of the article.

**Summary and General Comments**

Reviewer #1: MOOC based training on Implementation Research in health care is a novel approach to expand knowledge of implementation research. Building capacity of public health researchers and managers, policy and program managers in Implementation Research is the need of the hour across the developing countries.

Recommended for publication

Reviewer #2: The article manuscript entitled “Improving understanding of disease control implementation research through a MOOC with participants from low- and middle-income countries” is a crucial contribution to the assessment of the effect of MOOCs in general, and in implementation research more specifically, and with LMICs in particular. The work should definitely made accessible to the wider audience of researchers, policy makers, and implementers in the PLoS NTD. The data is promising to show interesting results that can be used for the furthering of MOOCs and similar programmes.

However, the manuscripts needs a couple of clarifications and specifications to show what is intended to be shown.

Reviewer #3: It is a good text that describes the first experience of MOOC training in implementation research. It would probably be appropriate to report that a substantial part of the students came from countries where the subject has already been discussed and experimented with in recent years. The big challenge is to take the course to regions where IR is still little known, but it is just as necessary, if not more.

The language will likely remain a challenge. Public policies and the structuring of health systems are also very different, and solutions must be local. 

However, perhaps it is the beauty of IR: a continuous evolving process that needs to be revisited for each disease in each territory. Methodologies are essential as a guide to the path to be followed, but always need to be adapted to the lived world.

PLOS authors have the option to publish the peer review history of their article (what does this mean?). If published, this will include your full peer review and any attached files.

Reviewer #1: Yes: Shiv Dutt Gupta

Reviewer #2: Yes: Dr Michael Kaser

Reviewer #3: No
---

## [Decision Letter · Decision Letter 1]

28 Nov 2022

Dear Dr Ross,

Thank you very much for submitting your manuscript "Improving Understanding of Disease Control Implementation Research through a MOOC with Participants from Low- and Middle-Income Countries: Evaluating Participant Reactions and Learning" for consideration at PLOS Neglected Tropical Diseases. As with all papers reviewed by the journal, your manuscript was reviewed by members of the editorial board and by several independent reviewers. The reviewers appreciated the attention to an important topic. Based on the reviews, we are likely to accept this manuscript for publication, providing that you modify the manuscript according to the review recommendations. 

Sincerely,

Alberto Novaes Ramos Jr

Academic Editor

Victoria Brookes

Section Editor

Reviewer's Responses to Questions

**Key Review Criteria Required for Acceptance?**

**Methods**

-Are the objectives of the study clearly articulated with a clear testable hypothesis stated?

-Is the study design appropriate to address the stated objectives?

-Is the population clearly described and appropriate for the hypothesis being tested?

-Is the sample size sufficient to ensure adequate power to address the hypothesis being tested?

-Were correct statistical analysis used to support conclusions?

-Are there concerns about ethical or regulatory requirements being met?

Reviewer #2: (No Response)

Reviewer #3: Overall, the text became more evident after the review. As previously pointed out, it is an exploratory study; therefore, its innovative character and the importance of the topic discussion justify any minor limitations.

**Results**

-Does the analysis presented match the analysis plan?

-Are the results clearly and completely presented?

-Are the figures (Tables, Images) of sufficient quality for clarity?

Reviewer #2: (No Response)

Reviewer #3: All of the items mentioned above have positively been achieved.

**Conclusions**

-Are the conclusions supported by the data presented?

-Are the limitations of analysis clearly described?

-Do the authors discuss how these data can be helpful to advance our understanding of the topic under study?

-Is public health relevance addressed?

Reviewer #2: (No Response)

Reviewer #3: All of the items mentioned above have positively been achieved.

**Editorial and Data Presentation Modifications?**

Reviewer #2: (No Response)

Reviewer #3: It should be best explained the number of MOOC completers throughout the text: 1163 (Results)/546 (Table 1)/438(p.15). Indeed, what profile is the most important to authors: the completers or the survey participants?

Attention to page 18 when it says that the new languages versions will be available at the beginning of 2020/2021.

**Summary and General Comments**

Reviewer #2: Review of revised version

The article manuscript entitled “Improving understanding of disease control implementation research through a MOOC with participants from low- and middle-income countries: evaluating participant reactions and learning” continuous to give a crucial contribution to experiences with MOOCs for teaching and training purposes towards the research and implementation community, in particular in LMICs.

Some of the reviewer’s comments were addressed, others not or addressed half-heartedly. In principle the manuscript should be published but it is up to the editor to which extend reviewers investment on suggestions should be implemented before a final decision.

The revised manuscript was re-sent to the reviewer. Unfortunately, the line numbering is not any more provided.

Some of the highlighted points were addressed:

- the title has been made more precise

- elements have been reassigned to the sections where the reader expects them (introduction, methods, results, discussions)

- the overall readability has improved

- the authors included a table as suggested which helps already 

This table should, however, be improved – this table could stimulate the reader much more when (i) harmonized, i.e. always the same order (public health researchers, students,…) rather than randomly listed, (ii) information from the text could go into it, i.e. page 13, paragraph starting with “The highest education level”. If the information becomes too complex for one table, the content can be split into information for two different tables, some reflections on this could help improve and reduce the body to the text.

Some of the other mentioned points were not addressed and lead to dissatisfaction:

- the methods are still scare and could be more detailed as the methods section should describe exactly so that the study could be repeated by anyone. However, some of the methodology is still scattered in other sections: i.e. page 7 paragraph starting with “The MOOC was hosted…” which is in introduction. If there was no free-text option in the survey, it should be mentioned, otherwise it would be an asset if some key citations could be added from these free texts.

- In particular introduction and discussion are still longish, show redundancies and are hence still a bit cumbersome to read, also page 14 paragraph starting with “The highest levels” is redundant to other passages. The beginning of the discussion is results and should go there.

Minor points:

- the title: it should most likely be plural to read: “evaluating participant reactions and learnings” ?

- some of the (unclear or misleading) formulations or imprecise wordings pointed out in the original review remained untouched, as are harmonization in order of listing (i.e. some of the “bachelor, master, PhD” listings).

- page 13 bottom: change to read “other BRICS countries”

Reviewer #3: (No Response)

PLOS authors have the option to publish the peer review history of their article (what does this mean?). If published, this will include your full peer review and any attached files.

Reviewer #2: No

Reviewer #3: No

Figure Files:

Data Requirements:

Reproducibility:

References

---

## [Decision Letter · Decision Letter 2]

5 Feb 2023

Dear Dr Ross,

We are pleased to inform you that your manuscript 'Improving Understanding of Disease Control Implementation Research through a MOOC with Participants from Low- and Middle-Income Countries: Evaluating Participant Reactions and Learning' has been provisionally accepted for publication in PLOS Neglected Tropical Diseases.

Best regards,

Alberto Novaes Ramos Jr

Academic Editor

Victoria Brookes

Section Editor

Reviewer's Responses to Questions

**Key Review Criteria Required for Acceptance?**

**Methods**

-Are the objectives of the study clearly articulated with a clear testable hypothesis stated?

-Is the study design appropriate to address the stated objectives?

-Is the population clearly described and appropriate for the hypothesis being tested?

-Is the sample size sufficient to ensure adequate power to address the hypothesis being tested?

-Were correct statistical analysis used to support conclusions?

-Are there concerns about ethical or regulatory requirements being met?

Reviewer #2: (No Response)

Reviewer #3: (No Response)

**Results**

-Does the analysis presented match the analysis plan?

-Are the results clearly and completely presented?

-Are the figures (Tables, Images) of sufficient quality for clarity?

Reviewer #2: (No Response)

Reviewer #3: (No Response)

**Conclusions**

-Are the conclusions supported by the data presented?

-Are the limitations of analysis clearly described?

-Do the authors discuss how these data can be helpful to advance our understanding of the topic under study?

-Is public health relevance addressed?

Reviewer #2: (No Response)

Reviewer #3: (No Response)

**Editorial and Data Presentation Modifications?**

Reviewer #2: (No Response)

Reviewer #3: (No Response)

**Summary and General Comments**

Reviewer #2: (No Response)

Reviewer #3: (No Response)

PLOS authors have the option to publish the peer review history of their article (what does this mean?). If published, this will include your full peer review and any attached files.

Reviewer #2: No

Reviewer #3: No

---

## [Editor Report · Acceptance letter]

20 Mar 2023

Dear Dr Ross,

We are delighted to inform you that your manuscript, "Improving Understanding of Disease Control Implementation Research through a MOOC with Participants from Low- and Middle-Income Countries: Evaluating Participant Reactions and Learning," has been formally accepted for publication in PLOS Neglected Tropical Diseases.

Best regards,

Shaden Kamhawi

co-Editor-in-Chief

Paul Brindley

co-Editor-in-Chief
